# Total synthesis of cyrneines A–B and glaucopine C

Guo-Jie Wu[1], Yuan-He Zhang[1,2], Dong-Xing Tan[1,2] & Fu-She Han ⬤ [1,3]

The cyrneine diterpenoids represent a structurally intriguing subfamily of cyathane diterpenoids and could significantly induce neurite outgrowth. Therefore, the efficient synthesis of these natural products is of great importance. Herein, we present a route for the collective synthesis of cyrneines A, B, and glaucopine C. As the key precursor, the 5-6-6-tricyclic scaffold is efficiently constructed by employing a mild Suzuki coupling of heavily substituted nonactivated cyclopentenyl triflate and a chelation-controlled regiospecific Friedel-Crafts cyclization as key transformations. The stereoselective installation of the all-carbon quaternary center at $C_6$ ring junction of the tricycle is achieved via Birch reductive methylation. Subsequently, a carbenoid-mediated ring expansion furnishes the essential 5-6-7-tricyclic core. Finally, manipulation of this core by several appropriately orchestrated conversions accomplishes a more step-economic synthesis of cyrneine A (20 steps), and the first synthesis of cyrneine B (24 steps) and glaucopine C (23 steps).

[1] Key Lab of Synthetic Rubber, Changchun Institute of Applied Chemistry, Chinese Academy of Sciences, 5625 Renmin Street, Changchun, Jilin 130022, China. [2] University of Science and Technology of China, Hefei, Anhui 230026, China. [3] State Key Lab of Fine Chemicals, Dalian University of Technology, Dalian 116024, China. Correspondence and requests for materials should be addressed to F.-S.H. (email: fshan@ciac.ac.cn)

The cyathane diterpenoids compose a large family of natural products with greater than 100 members[1–3]. These molecules feature a common 5-6-7 fused tricarbocyclic core with most of which possessing two all-carbon quaternary stereocenters at the ring junctions with anti-orientation. The unique structural framework coupled with the diverse oxidation and unsaturation states around the ring periphery gives rise to the structural complexity and diversity. Biological evaluation revealed that cyathane diterpenoids exhibit a rich variety of biological activities such as antibiotics, antimicrobial, antitumor, anti-inflammatory, and most significantly, nerve growth factor (NGF)-regulating properties. Over the roughly 20 years, considerable total synthesis efforts have been conducted, and a number of molecules within the subclasses of allocyathin, erinacine, sarcodonin, cyathin, scabronine, and cyanthiwigin have been synthesized in enantioselective or racemic forms. Of note are important contributions from the groups of Nakada[2], Ward[4,5], Trost[6,7], Danishefsky[8], Stoltz[9], Phillip[10], Reddy[11], Snider[12,13], and others[14,15].

The cyrneines A−E and glaucopine C[16-19], isolated form the *Sarcodon Cyrneus*, are a novel subfamily of cyathane diterpenoids (Fig. 1a). Unique to the structures of this subfamily as compared with other subclasses[2–15] is the extra oxidation at $C_1$ (e.g., 1, 3, 5, and 6), or at both $C_1$ and $C_4$ (e.g., 2 and 4) in the five-membered

ring. This causes considerable synthetic challenges resulting from the higher oxidation states as well as the increased stereocenters at $C_1$ or $C_4$, especially for compounds 2 and 4 bearing an allylic functionality at $C_4$ and two vicinal quaternary centers at $C_4$ and $C_9$ ring junction. To date, only cyrneine A (1) has been synthesized by Gademann and co-workers[20] with an elegant sequence of 24 steps from (-)-(R)-carvone (Fig. 1b). The key transformations involved a reductive Knoevenagel condensation, a Heck cyclization, and a Yamamoto ring expansion. On the other hand, biological evaluations showed the activity of this subfamily was markedly influenced by the minor structural differences. Among the molecules evaluated, cyrneines A (1) and B (2) could induce significantly the neurite outgrowth in PC12 cells and the expression of NGF in 1321N1 cells in a concentration-dependent manner. Thus, driven by the structural complexity, biological potential, and the interest for an in-depth SAR elucidation, it is of great importance to develop a strategically new route that could be used for efficient and versatile synthesis of these natural products and potential analogues. Herein, we report such a route as demonstrated by a more step-economic synthesis of cyrneine A (1), and the first synthesis of cyrneine B (2) and glaucopine C (3).

## Results

**Retrosynthetic analysis**. We envisioned that the 5-6-7 tricyclic core **A** could serve as an advanced intermediate for our divergent synthesis (Fig. 2). The key challenge for accessing cyrneine A (1) and the intermediate **B** would be the β-selective reduction of carbonyl at $C_{14}$ in **A**. The glaucopine C (3) and cyrneine B (2)

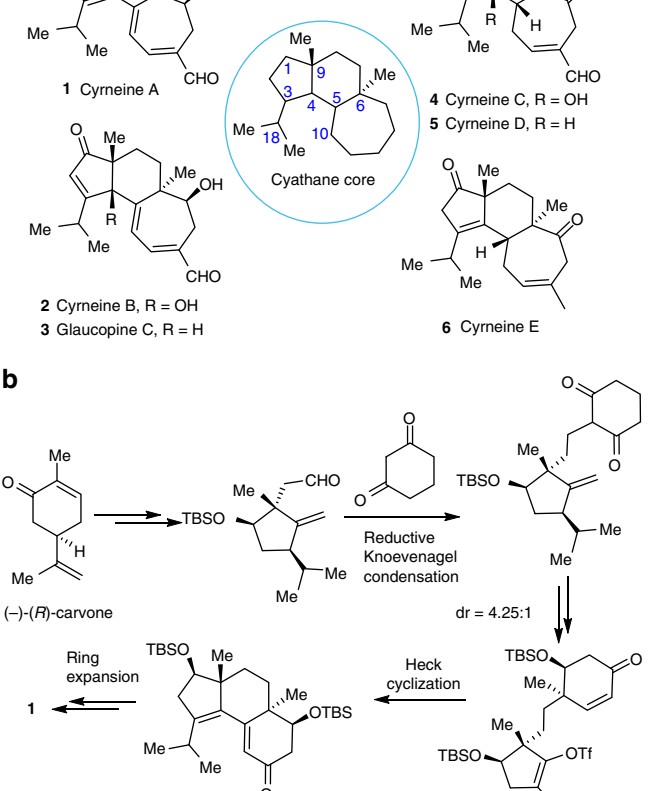

**Fig. 1** Cyathane diterpenoid natural products. **a** The structures of cyrneines A−E and glaucopine C. The [5.6.7]-tricycle highlighted in blue circle indicates a common core structural motif shared by the natural products. **b** The key reactions included in Gademann's synthesis of cyrneine A. Me methyl, Tf trifluoromethanesulfonyl, TBS *tertiary*-butyldimethylsilyl, dr diastereomeric ratio

**Fig. 2** Retrosynthetic analysis of cyrneine A−B and glaucopine C. PG stands for protecting group; F–C represents Friedel–Crafts; unless otherwise noted, R in structures **E–J** indicates hydrogen or methyl group

were planned to be synthesized from **C** by means of 1,3-proto-tropic shift and γ-oxidation at $C_4$, albeit numerous concerns such as stereo- as well as regio- and chemoselectivity are appreciably involved in these late-stage manipulations arising from the conjugation effect of multiple olefinic functionalities in the 5- and 7-membered ring, and the acidity of proton at $C_2$ and $C_{18}$. The

intermediate **A** was proposed to be derived from the 5-6-6 tricyclic **D** by ring expansion. For the installation of the all-carbon quaternary stereocenter at the angular $C_6$, we conceived of Birch reductive methylation from tricyclic ketone **E**. To our knowledge, such transformation has never been explored in the synthesis of cyathane-type natural products presumably due to the concerns of annular strain, steric hindrance, and stereoselectivity[1–3]. However, the great potential to streamline the synthesis of cyathane compounds prompted us to investigate it. The construction of the crucial tricyclic system **F** might be achieved through an intramolecular Friedel-Crafts reaction of aldehyde **G** whose synthesis would be implemented through the Suzuki-Miyaura cross-coupling of a heavily substituted nonactivated vinyl triflate **H**. Simplification of **H** revealed the five-membered cyclic ketone **I**, which was to be prepared through an enantioselective desymmetric reduction of 2,2-disubstituted 1,3-cyclopentanedione **J**.

**Fig. 3** Synthesis of **15a**. Reagents and conditions: (a) **8** or **9** (2.0 equiv), $K_2CO_3$ (1.5 equiv), acetone, rt, overnight (75−80%); (b) Yeast extract (50 wt%), D-glucose, dry active baker's yeast, $H_2O$, rt, 36 h (67%); (c) p-$NO_2$-$C_6H_4CO_2H$ (2.0 equiv), $PPh_3$ (2.0 equiv), DEAD (2.0 equiv), THF, 0 to 50 °C, overnight (90%); (d) $K_2CO_3$ (2.0 equiv), MeOH/THF (v/v = 1:2), 0 °C, 30 min (98%); (e) TBSCl (1.2 equiv), imidazole (1.2 equiv), DMF, rt, overnight (91%); (f) NaH (5.0 equiv), iPrI (10.0 equiv), THF, reflux, overnight; then aq. HCl (2 M), rt, 1.5 h (88%); (g) LiHMDS (1.3 equiv, 1.0 M in THF), PhNTf₂ (1.3 equiv), THF, −78 °C to rt, 3 h (80%). DEAD diethyl azodicarboxylate, TBSCl t-butyldimethylsilyl chloride, LiHMDS lithium hexamethyldisilazide, THF tetrahydrofuran, DMF N,N-dimethylformamide

**Synthesis of cyrneine A**. The synthesis commenced with asymmetric synthesis of the cyclopentenyl triflate **15a** (Fig. 3). Accordingly, allylation of the readily available 2-methyl 1,3-cyclopentanedione **7** with allylic bromide **8** or **9** afforded 2,2-disubstituted cyclopentanedione **10** and **11** in high yields, respectively. Desymmetric enantioselective reduction of **10** and **11** was carried out by CBS reduction[21,22]. However, the result was less satisfactory in terms of enantio- and diastereoselectivity, and scalability. We then inspected the enzyme-catalyzed reduction with baker's yeast[23,24]. Excellent enantioselectivity of up to 99% ee and moderate diastereoselectivity of ca. 8–9:1 were observed for **10**. The imperfect diastereoselectivity prompted us to investigate the sterically more hindered **11**. Delightedly, the d.r. ratio could be improved to ca. 25:1. Thus, a comparison of the results obtained from different ways showed that the enzyme-catalyzed reduction of prenyl substituted **11** afforded the best outcome. The reaction could be uneventfully performed on decagram scales to give α-hydroxyketone **12** in 65–68% yield with 99% ee and 25:1 d.r. (see Supplementary Fig. 36). Configuration inversion of the

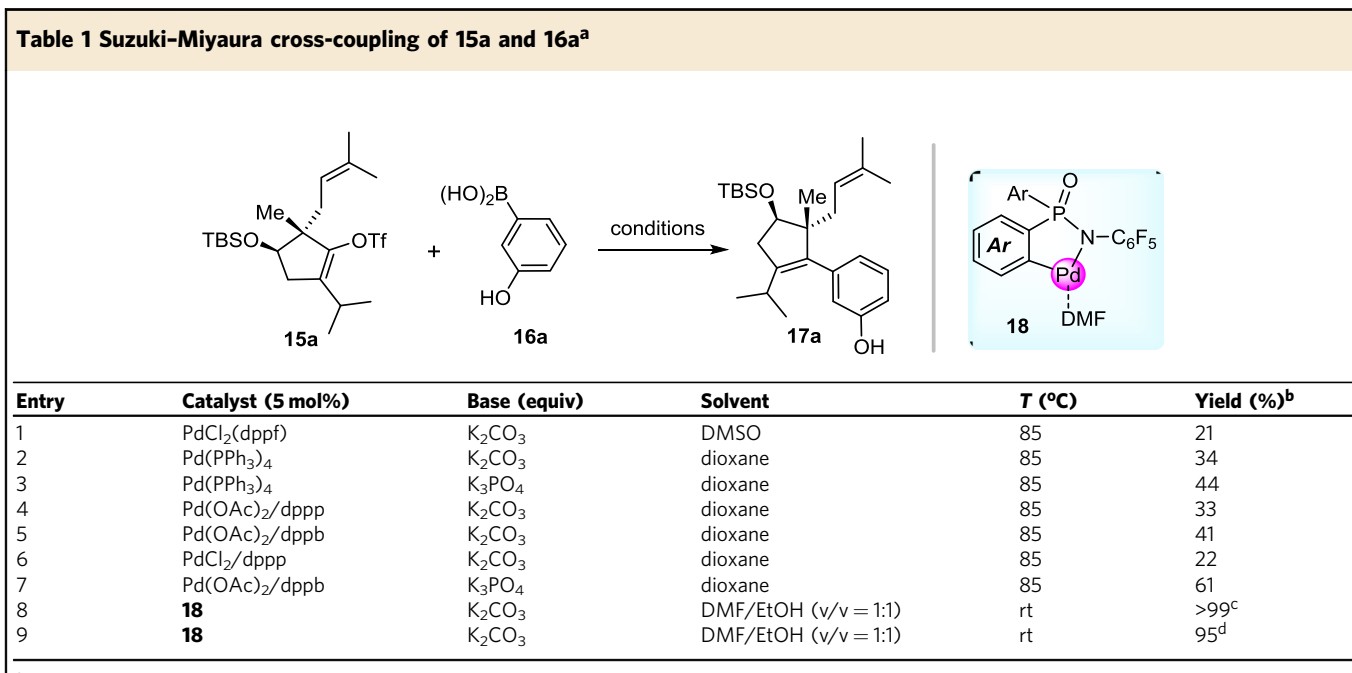

**Table 1 Suzuki–Miyaura cross-coupling of 15a and 16a[a]**

| Entry | Catalyst (5 mol%) | Base (equiv) | Solvent | T (ºC) | Yield (%)[b] |
|---|---|---|---|---|---|
| 1 | PdCl₂(dppf) | K₂CO₃ | DMSO | 85 | 21 |
| 2 | Pd(PPh₃)₄ | K₂CO₃ | dioxane | 85 | 34 |
| 3 | Pd(PPh₃)₄ | K₃PO₄ | dioxane | 85 | 44 |
| 4 | Pd(OAc)₂/dppp | K₂CO₃ | dioxane | 85 | 33 |
| 5 | Pd(OAc)₂/dppb | K₂CO₃ | dioxane | 85 | 41 |
| 6 | PdCl₂/dppp | K₂CO₃ | dioxane | 85 | 22 |
| 7 | Pd(OAc)₂/dppb | K₃PO₄ | dioxane | 85 | 61 |
| 8 | 18 | K₂CO₃ | DMF/EtOH (v/v = 1:1) | rt | >99[c] |
| 9 | 18 | K₂CO₃ | DMF/EtOH (v/v = 1:1) | rt | 95[d] |

[a]Reaction conditions: **15a** (47.0 mg, 0.1 mmol), **16a** (27.6 mg, 0.2 mmol), catalyst (5 mol%), and base (3.0 equiv), in 1 mL solvent under nitrogen atmosphere for 12 h
[b]Isolated yield
[c]The reaction was run at rt for 5 h
[d]The reaction was performed with 6.6 g of **15a** and was quenched until **15a** had disappeared as monitored by TLC

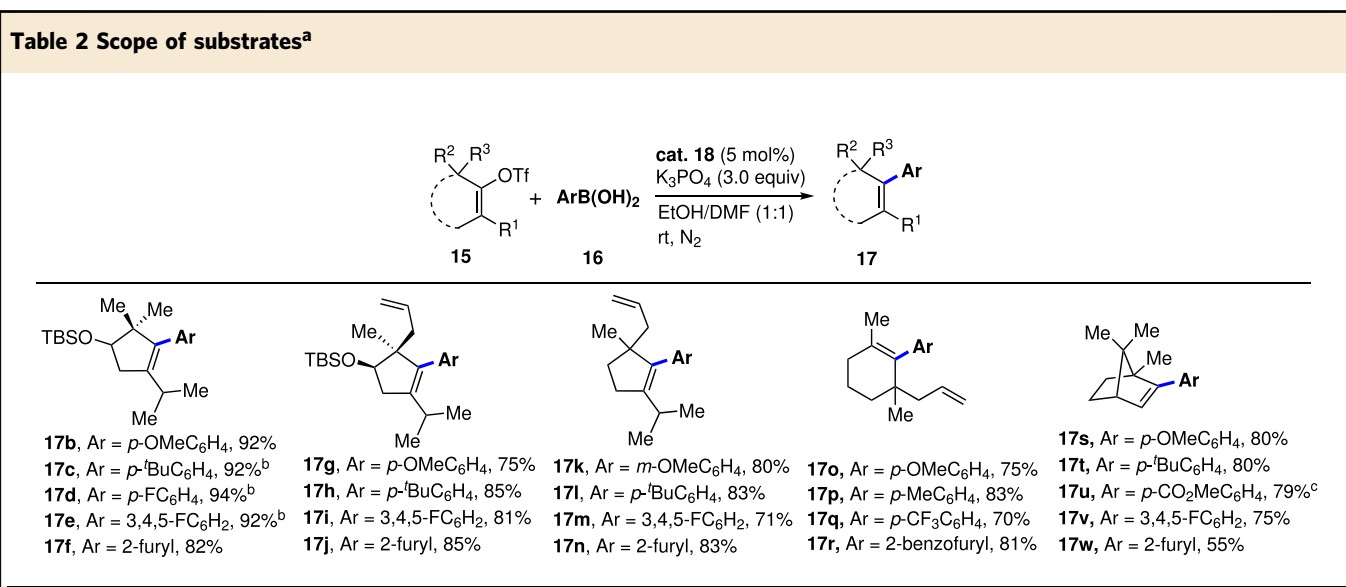

**Table 2 Scope of substrates[a]**

**17b**, Ar = p-OMeC₆H₄, 92%
**17c**, Ar = p-ᵗBuC₆H₄, 92%[b]
**17d**, Ar = p-FC₆H₄, 94%[b]
**17e**, Ar = 3,4,5-FC₆H₂, 92%[b]
**17f**, Ar = 2-furyl, 82%

**17g**, Ar = p-OMeC₆H₄, 75%
**17h**, Ar = p-ᵗBuC₆H₄, 85%
**17i**, Ar = 3,4,5-FC₆H₂, 81%
**17j**, Ar = 2-furyl, 85%

**17k**, Ar = m-OMeC₆H₄, 80%
**17l**, Ar = p-ᵗBuC₆H₄, 83%
**17m**, Ar = 3,4,5-FC₆H₂, 71%
**17n**, Ar = 2-furyl, 83%

**17o**, Ar = p-OMeC₆H₄, 75%
**17p**, Ar = p-MeC₆H₄, 83%
**17q**, Ar = p-CF₃C₆H₄, 70%
**17r**, Ar = 2-benzofuryl, 81%

**17s**, Ar = p-OMeC₆H₄, 80%
**17t**, Ar = p-ᵗBuC₆H₄, 80%
**17u**, Ar = p-CO₂MeC₆H₄, 79%[c]
**17v**, Ar = 3,4,5-FC₆H₂, 75%
**17w**, Ar = 2-furyl, 55%

[a]Conditions: Enol triflate **15** (0.2 mmol), boronic acid **16** (0.4 mmol), K₃PO₄ (3.0 equiv) in a mixed EtOH/DMF (v/v = 1:1) solvent (2.0 mL) at room temperature under nitrogen. Isolated yield. The use of K₃PO₄ instead of K₂CO₃ was more effective for the coupling of boronic acids without free phenol group
[b]The yield was determined based on the ¹H NMR spectroscopy because of the contamination of a small amount of inseparable homocoupled product of boronic acid
[c]The methyl ester group of the product was partially exchanged to ethyl ester

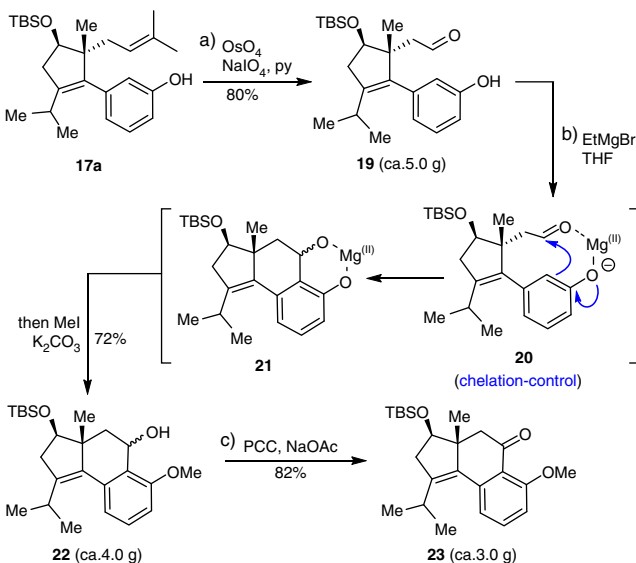

**Fig. 4** Synthesis of the tricyclic compound **23**. Reagents and conditions: (a) OsO₄ (4 mol%), NaIO₄ (5.0 equiv), pyridine (3.0 equiv), dioxane/H₂O (v/v = 5:1), 80 °C (80%); (b) EtMgBr (1.1 eqiuv, 1.0 M in THF), −78 to 40 °C, overnight, then K₂CO₃ (2.0 equiv), MeI (5.0 equiv), and DMF were charged in situ, 55 °C, 10 h (72%); (c) PCC (4.0 equiv), NaOAc (4.0 equiv), celite (ca. 130 wt%), rt, 6 h (82%). THF tetrahydrofuran, DMF N,N-dimethylformamide, PCC pyridinium chlorochromate

α-hydroxy group in **12** was carried out smoothly under Mitsunobu conditions to give the desired β-hydroxy intermediate **13**. Protection of the β-hydroxy in **13** with TBS followed by a base-mediated isopropylation with *i*PrI provided ketone **14** in 80% yield over two steps (94% brsm). A literature survey showed that α-alkylation of ketone with secondary alkyl iodide was relatively difficult[25,26]. We examined the isopropylation under the effect of LDA, MHMDS (M = Li, Na, K), and NaH, respectively. The

results showed that NaH was a more efficient base. Treatment of **14** with LiHMDS and PhNTf₂ afforded the enol triflate **15a**.

The Suzuki-Miyaura cross-coupling involving sterically congested nonactivated enolate substrates was relatively rare. Initial scouting of the conditions for coupling triflate **15a** with **16a** was carried out by examining an array of conditions reported in literature[27–31] (Table 1). Unfortunately, most of the reactions were less effective, affording the coupled product **17a** in low yields (entries 1−6). While a combination of Pd(OAc)₂ and dppb ligand could afford the product in moderate yield (entry 7), a large scale synthesis was problematic owing to an elongated reaction time, leading to partial hydrolysis of **15a**. To compare with the prior literature[27–31], the low coupling efficiency is probably due to the excessively congested structure of **15a** with the additional presence of a bulky OTBS. To overcome this obstacle, we turned to examine the catalytic efficiency of phosphinamide-derived palladacycle **18**, which was developed in our previous studies toward synthesizing P-stereogenic compounds through C–H arylation of phosphinamide[32,33] and exhibited excellent catalytic activity for Suzuki cross-coupling of aryl (pseudo)halides under mild conditions[34]. To our delight, we found that the palladacycle **18** did display high catalytic activity for such a heavily substituted nonactivated triflate **15a**. The coupled product **17a** could be obtained in almost quantitative yield at room temperature (entry 8). Most significantly, the reaction could be reliably scaled up to multigram scales (6.6 g of **15a**) without compromising the yield (entry 9).

To further expand the potential utility of the new precatalyst for mild and effective coupling of sterically hindered nonactivated enolate derivatives, we examined the substrate scope by varying the structures of both reaction partners. Effective cross-coupling was observed for an extensive combination of an array of heavily substituted nonactivated enol triflates and aryl boronic acids. As shown in Table 2, both the five-membered (**17b**–**17n**) and six-membered (**17o**–**17r**) cyclic enol triflates reacted smoothly with a rich range of aryl boronic acids whose structure was modified by electron-donating OMe, *t*Bu, and Me, as well as electron-withdrawing F, 3,4,5-trifluoro, CF₃, and CO₂Me groups. In addition, a [2.2.1]-bridged bicycle (**17s**–**17w**) also exhibited good

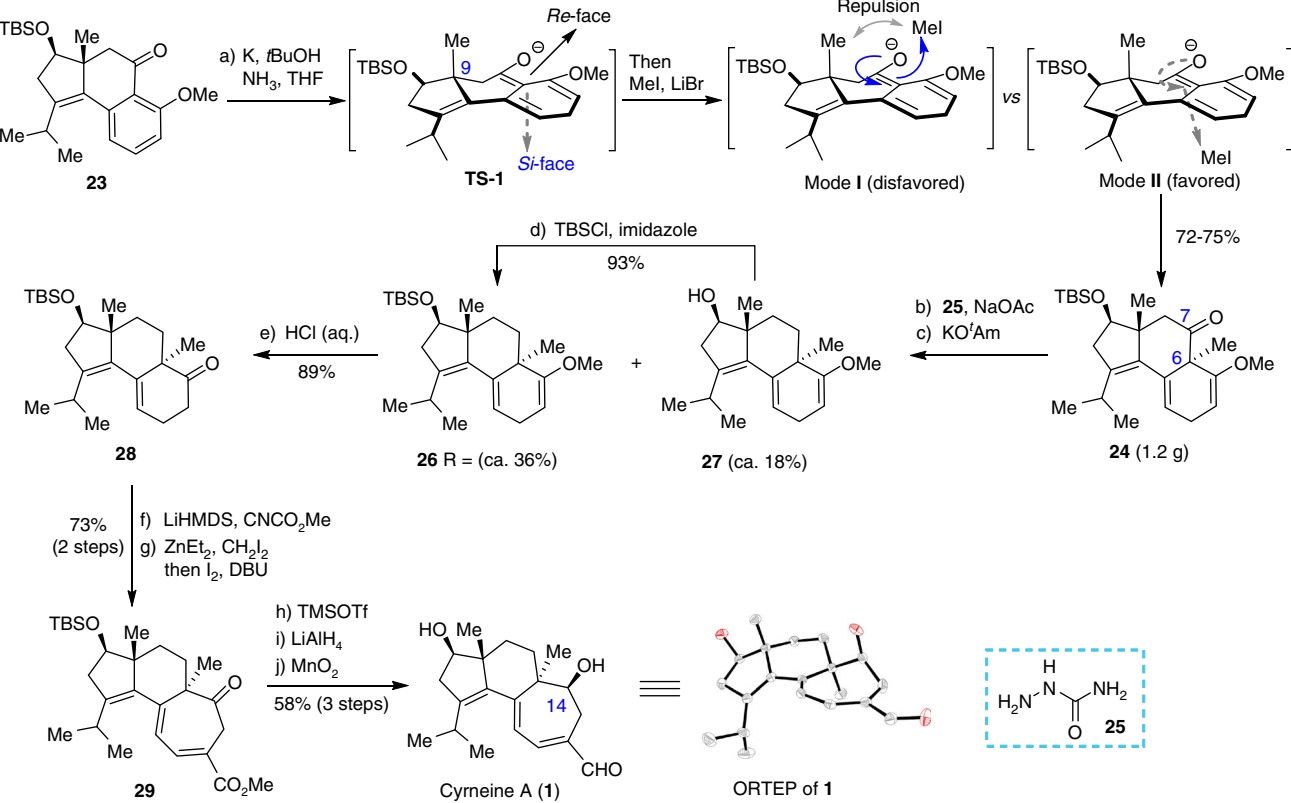

**Fig. 5** The completion of the synthesis of cyrneine A. Reagents and conditions: (a) Liq. NH₃, *t*BuOH (1.0 equiv), K (2.5 equiv), Et₂O, −78 °C, 10 min; then LiBr (2.5 equiv), MeI (5.0 equiv), and THF were charged in situ, −78 °C, 1 h, then warmed to rt over a period of 1 h (72%); (b) **25** (5.0 equiv), NaOAc (5.0 equiv), EtOH, 35 °C, 3 h; (c) KO*t*Am (5.0 equiv), degassed xylene, 140 °C, 1.8 h (36% for **26**; 18% for **27**); (d) TBSCl (1.2 equiv), imidazole (1.2 equiv), DMF, rt, overnight (93%); (e) 5% aq. HCl, THF, 15 to 20 °C, 40 min (89%); (f) LiHMDS (1 M in THF, 3.5 equiv), CNCO₂Me (3.0 equiv), THF, −78 °C, 1 h; (g) ZnEt₂ (1.0 M in hexane, 6.0 equiv), CH₂I₂ (6.0 equiv), CH₂Cl₂, rt, 2 h; I₂ (7.0 equiv), rt; then saturated aq. Na₂S₂O₃ (excess), rt; then DBU (20.0 equiv), rt, 10 min (73% for 2 steps); (h) TMSOTf (5.0 equiv), CH₂Cl₂, -25 °C, 15 min; (i) LiAlH₄ (6.0 equiv), Et₂O, 10 min at −78 °C, then warmed to rt; (j) MnO₂ (50 equiv), CH₂Cl₂, rt, overnight (58% in 3 steps); *t*Am = *tert*-amyl; DBU = 1,8-diazabicyclo[5.4.0]undec-7-ene; TMSOTf = trimethylsilyl trifluoromethanesulfonate

compatibility. Notably, the reaction was viable for 2-furyl and benzo-2-furyl boronic acids (**17f**, **17j**, **17n**, **17r**, and **17w**). These results would be appealing because coupling of 2-heteroaryl boronic acids is a considerable challenge resulting from the detrimental pyrolysis under conventional heating conditions[35], especially with the nonactivated enol triflates as seen from the sporadic examples explored in the synthesis of relevant natural products[36].

Having established the robust conditions for large scale synthesis of **17a** in this way, we then shifted our focus to construct the 5-6-6 tricarbocyclic system **23** (Fig. 4). Oxidation of the double bond in **17a** delivered the aldehyde **19**. For the Friedel-Crafts cyclization of **19**, we first examined the acid-mediated protocols as investigated in Trost's[28] and Jiang's[37] synthesis of hamigeran B. However, similar to their outcome, the reaction proved to be futile owing to multiple difficulties associated with the poor regioselectivity of *para* vs. *ortho* to phenolic OH, the lability of OTBS group, and the ease of further dehydration of the cyclized product. After a careful deliberation, we devised a chelation-controlled strategy and found that EtMgBr was a suitable reagent. Namely, treatment of **19** with EtMgBr produced a magnesium phenolate salt. The Mg(II) ion may be acting to serve as a Lewis acid to coordinate with the aldehyde group, and thereby forming intermediate **20** through chelation-control. Consequently, the Friedel-Crafts reaction proceeded exclusively at the position *ortho* to phenolic OH to afford the tricyclic product **21**. Upon in situ selective methylation of

phenolic OH, the tricyclic alcohol **22** could be obtained as a single regioisomer in 64–72% yield on multigram scales. Oxidation of **22** gave the corresponding ketone **23** smoothly.

Next, our task was moved forward to complete the synthesis of cyrneine A (**1**) (Fig. 5). Attempted installation of methyl group at C₆ in **23** was carried out by executing the Birch reductive methylation. However, initial trials showed that the reaction was considerably challenging. Only a complex mixture was afforded under a broad array of conditions. Based on the NMR analysis of the crude products, the reaction was mainly complicated by the competitive reduction of C₃ = C₄ double bond, aryl ring, and carbonyl group without observation of the methylated product. After an exhaustive screening and optimization of conditions, we could successfully install the angular methyl group based on the methods reported early by Narisada[38] and Gibbard[39]. It was found that addition of scrupulously dried LiBr was crucial. The desired product **24** could be obtained in 72–75% yield as a single diastereoisomer over gram scale. The exceptionally high stereo-selectivity is presumably attributed to the steric hindrance of methyl group at C₉, which compels the nucleophilic attack of **TS-1** anion to MeI to take place from the less hindered *si*-face as illustrated by mode I versus mode II. The stereochemistry of **24** was determined by NOESY correlations (see Supplementary Fig. 21) and was further confirmed by X-ray single crystal analysis of the final product (vide infra).

Concerning the reduction of the carbonyl functionality in the central ring of **24**, many unexpected obstacles were encountered.

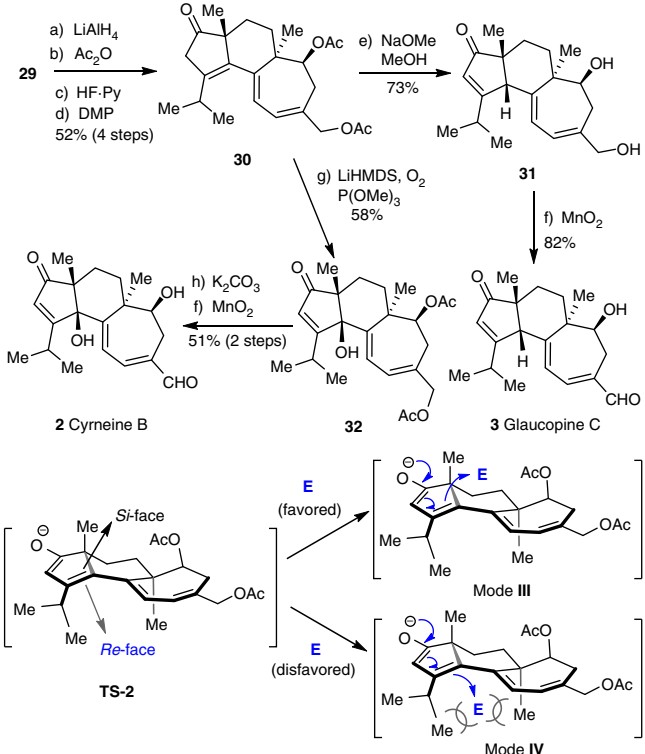

**Fig. 6** The synthesis of cyrneine B and glaucopine C. Reagents and conditions: (a) LiAlH$_4$ (6.0 equiv), Et$_2$O, 20 min at −78 °C, then warmed to rt; (b) Ac$_2$O (15.0 equiv), DMAP (20 mol%), pyridine, 30 min, rt; (c) HF·Py (excess), THF, rt, 2 h; (d) DMP (1.6 equiv), CH$_2$Cl$_2$, 30 min, rt (52% for 4 steps); (e) NaOMe, MeOH, 35 °C, 4 h (73%); (f) MnO$_2$ (15 equiv), CH$_2$Cl$_2$, rt, 3 h (82%); (g) LiHMDS (2.0 equiv), THF, −78 °C, 15 min, then P (OMe)$_3$ (4.0 equiv), O$_2$, 2 h (58%); (h) K$_2$CO$_3$ (excess), MeOH, 35 °C, 4 h (51% for 2 steps). DMAP 4-(dimethylamino)pyridine, Py pyridine, DMP Dess–Martin periodinane

First, we examined the Barton-McCombie[40] radical reduction involving reduction of the carbonyl group (NaBH$_4$, MeOH), thiocarbonation of hydroxy (NaH, CS$_2$, MeI), and radical induced deoxygenation. However, the product **26** was obtained in lower than 20% yield under a free combination of various radical initiators (e.g., AIBN, ABCN, Et$_3$B,) and reductants (e.g., Bu$_3$SnH, TMSH, TTMSS, and Ph$_2$SiH$_2$). Although no vigorous proof, it seemed that the cleavage of C$_6$−C$_7$ bond occurred facilely under radical conditions. Alternatively, the Wolff–Kishner–Huang type reduction was also proved to be ineffective under an array of routine conditions[41–48]. Extensive decomposition of substrate was observed. Fortunately, a patient investigation revealed that the reaction of **24** with semicarbazide **25**[49,50] followed by treatment of the resulting semicarbazone with KO$t$Am in degassed xylene afforded **26** in ca. 36% yield accompanied by ca. 18% of desilylated **27**, which could be resilylated to give **26**. As a result, **26** could be obtained in 53% yield in a synthetically useful level. Hydrolysis of the vinyl ether in **26** gave ketone **28**, which was immediately subjected to the ring expansion reaction without careful purification since **28** was somewhat easily oxidized under ambient atmosphere. For ring expansion, Nakada in the synthesis of allocyathin B$_2$[2] employed a four-step sequence involving acylation, iodomethylation, SmI$_2$-promoted ring expansion, and LDA/I$_2$-meidated elimination. Inspired by a protocol of Zercher[51], we accomplished the conversions through a two-step operation involving acylation followed by a one-pot Zn carbenoid-mediated ring expansion and I$_2$-promoted elimination. Compound **29** was thus obtained

efficiently in 73% yield over two steps. Finally, elaboration of **29** by removal of TBS, simultaneous reduction of ketone and ester, and selective oxidation of allylic primary alcohol furnished the total synthesis of cyrneine A (**1**) in 58% yield over three steps. Notably, the reduction proceeded highly stereoselectively at C$_{14}$ to afford β-OH product. The structure of **1** was unambiguously confirmed by $^1$H- and $^{13}$C-NMR, HRMS, and single crystal X-ray (CCDC 1830226) analysis. The data and dextrorotary property of the final product matched well with those of the reported natural sample[16] (See Supplementary Table 1 and 2, and Supplementary Fig. 26).

**Synthesis of cyrneine B and glaucopine C.** Having successfully synthesized cyrneine A, we entered the final stage toward synthesizing cyrneine B (**2**) and glaucopine C (**3**) (Fig. 6). Based on our retrosynthetic design, the advanced intermediate **29** was converted into β,γ-enone **30** through four routine transformations. After an optimization of reaction sequence and conditions, we found that the 1,3-prototropic shift and concomitant deprotection of acetyl in **30** proceeded smoothly under the effect of NaOMe to deliver the thermodynamically preferred enone **31** as a single C$_4$ β-H stereoisomer. The stereochemistry was determined by NOESY correlation (see Supplementary Fig. 29). The exceptionally good stereoselectivity is presumably resulted from the differential steric hindrance between re-face and si-face of **TS-2**. As a result, the enolate anion prefers to approach the electrophiles from the less congested si-face rather than the hindered re-face (i.e., mode **III** vs. **IV**). Subsequently, selective oxidation of the allylic primary alcohol completed the synthesis of glaucopine C (**3**). Encouraged by the highly stereo- and regioselective prototropic shift, the installation of C$_4$ β-hydroxy toward synthesizing cyrneine B (**2**) was carried out though a base-mediated prototropic shift and aerobic γ-CH oxidation cascade from **30**. While a complex mixture with high polarity was obtained in initial scouting of the conditions under the effect of weak bases and heating presumably due to over oxidation of olefins and acidic CH at C$_2$ and C$_{18}$, treatment of **30** with LiHMDS by lowering the temperature to −78 °C under O$_2$ atmosphere afforded **32** in 58% yield. Finally, removal of acetyl followed by selective oxidation of the allylic alcohol delivered cyrneine B (**2**). The structures of cyrneine B and glaucopine C were confirmed by various spectroscopic analyses and by comparison of the analytical data of the synthetic samples with those reported for natural products[16,19] (see Supplementary Table 3–6, and Supplementary Fig. 30–35).

## Discussion

In summary, we have developed an efficient route that allowed for a more step-economic total synthesis of cyrneine A (20 steps), and the first total synthesis of cyrneine B (24 steps) and glaucopine C (23 steps) from readily available commercial materials. The synthesis features the use of a mild and efficient Suzuki-coupling of heavily substituted nonactivated vinyl triflate, a Mg(II)-mediated chelation-controlled Friedel–Crafts cyclization, a Birch reductive methylation, and a zinc carbenoid-mediated ring expansion. These key transformations enable an efficient and rapid construction of the 5-6-7 tricyclic core. Finally, the divergent synthesis of the three natural products from this core is achieved through several carefully orchestrated manipulations involving a stereoselective reduction of C$_{14}$ carbonyl, a stereoselective 1,3-prototropic shift, and a stereo- and chemoselective prototropic shift/γ-CH oxidation cascade at C$_4$. The utility of the new strategy and methods for efficient and flexible synthesis of other structurally relevant products is currently underway.

## Methods

**Procedures for the synthesis of cyrneines A–B and glaucopine C**. The detailed experimental procedures for the synthesis of cyrneines A, B, and glaucopine C were provided in Supplementary Information.

**General procedure for the cross-coupling reaction of enol triflates**. To a solution of vinyl triflate **15** (0.2 mmol) in a mixed solvent of DMF (1.0 mL) and EtOH (1.0 mL) was added arylboronic acid **16** (0.4 mmol, 2.0 equiv), palladacycle **18** (5.6 mg, 5 mmol%), and $K_3PO_4$ [127 mg, 0.6 mmol, 3.0 equiv ($K_2CO_3$ was used for the coupling of **15a** with **16a**)] at room temperature under nitrogen atmosphere. The resulting mixture was stirred at the same temperature until the vinyl triflate had disappeared as monitored by TLC. The reaction mixture was then poured into water and extracted with ethyl acetate ($3 \times 25$ mL). The organic layer was combined, washed with brine, dried over $Na_2SO_4$, and concentrated under vacuum. The residue was purified by silica gel column chromatography to afford the desired cross-coupling product **17**.

**Data availability**. Chemical compound information including NMR and HRMS data, copies of $^1H$- and $^{13}C$-NMR of all new compounds, 2D NMR of compounds **24**, **31**, cyrneine B (**2**), and glaucopine (**3**), HPLC charts for compound **12**, and single X-ray crystal data of cyrneine A (**1**). This material is provided in Supplementary Information. The X-ray crystallographic coordinates for cyrneine A (**1**) reported in this study have also been deposited at the Cambridge Crystallographic Data Centre (CCDC), under deposition number 1830226. These data can be obtained free of charge from The Cambridge Crystallographic Data Centre via www.ccdc.cam.ac.uk/data_request/cif. Other supporting data related to this work are available from the corresponding author upon reasonable request.

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

## Acknowledgements

We thank Prof. Bao-Min Wang at Dalian University of Technology for helpful discussions on the revision of the manuscript. Financial support from NSFC (21572215, 21602215), and State Key Laboratory of Fine Chemicals (KF 1515) is acknowledged.

## Author contributions

F.S.H. conceived the synthetic strategy, directed the project, and wrote the manuscript. F.S.H. and G.J.W. analyzed the experiment results. G.J.W., Y.H.Z. and D.X.T. conducted the experimental work.

## Additional information

**Competing interests:** The authors declare no competing interests.

