## [Peer Review File · Nature Communications]

Reviewer #1 (Remarks to the Author):

The cyrneine diterpenoids are attractive targets for chemical synthesis. The densely substituted seven-membered ring of these molecules poses a considerable challenge. A number of interesting strategies have been developed, including Gademann's Heck annulation/ring expansion approach. The syntheses described in this paper were beautifully designed and executed.

The preparation of the cyclopentane building block is clever and efficient. The desymmetrizing reduction using baker's yeast gave excellent ee and dr. It should be noted that conventional CBS and Ru-based asymmetric reduction reactions usually lead to modest dr. The subsequent Suzuki–Miyaura coupling looked formidable. The authors elegantly solved this problem using their own phosphinamide-based palladacycle catalyst. The substrate scope of this palladacycle-catalyzed coupling reaction was studied as well. A 5,6,6-tricyclic system was then constructed through Friedel–Crafts reaction. One-pot Birch reduction/methylation introduced the quaternary C6. The Zn carbenoid-mediated ring expansion is clever. Finally, two highly oxidized molecules were conquered for the first time, and cyrneine A was prepared through a more interesting and efficient route.

I would like to emphasize that none of the steps is trivial to a total synthesis expert. The chemistry described here, e.g. asymmetric reduction, cross coupling, Birch/alkylation, modified Wolff–Kishner reduction, would certainly inspire the readers especially those in the field of natural product synthesis. The showcase of these reactions with highly demanding substrates provide a lot of useful information to the peers. The strategies developed in the syntheses are noteworthy as well. Although a number of groups contributed nicely to the construction of the two quaternary centers on the six-membered ring, the strategies presented in this paper is highly efficient and scalable and generally applicable.

Overall, from the method and strategy perspectives, this piece of work is interesting and outstanding, which should be accepted for publication in Nature Communications.

Reviewer #2 (Remarks to the Author):

This paper described the synthesis of a family of natural products, the cyrneines. The congestion in these molecules posed a challenge for their syntheses, but several interesting and effective solutions were devised for solving those problems. Firstly, the arylation by a Suzuki–Miyaura coupling of a very hindered enol triflate was accomplished, ultimately in excellent yield and multigram scale through

catalysis by complex 18. Then, the Friedel-Crafts cyclization at the ortho-position was accomplished by a chelation strategy. An alkylative Birch reduction installed the angular methyl group with the correct stereochemistry, and a challenging deoxygenation was overcome. Finally, a late-stage cyclopropanation-ring expansion generated the cycloheptadienone moiety. Several endgames provided cyrneine A, B, and glaucopine C. The quality of these syntheses merit publication in Nature Comm.

However, revisions are needed. Importantly, the rationalization of the stereoselective syntheses of 24 and 32 (TS1-4) must be corrected. Showing the anions as "lobes" does not correctly portray the species which should be enolates. For example, TS1-1 being disfavoured due to the hindrance between the methyl group and the anion is simply incorrect. The steric interactions should be with the incoming electrophile and the enolate. In addition, English errors should be corrected.

Reviewer #3 (Remarks to the Author):

Authors did not provide the checkcif pdf file for the present structure, although it is clearly stated in the guidelines for authors that "Prior to submitting any cif file, the structure factors and structural output must be checked using IUCr's CheckCIF routine at the <http://checkcif.iucr.org/> website. A pdf copy of the output supplied, explaining any A- or B level alerts should be also provided."

At the end of the comments, a list of issues with the submitted cif file is reported. Although there are no A and B-ALERTS, C and G-ALERTS were found, that could be easily addressed. The cif file could be accepted, once those issues are solved.

Comments:

1. The authors reported a Z value of 51 in the cif and res files. A Z value of 51 is not allowed for the orthorhombic space group P212121. In addition, in the Table of crystallographic data (page S45 of supporting information) the correct Z value of 4 is reported. The authors must perform another refinement cycle with the correct Z value. This would also prevent the ALERTS G (2-3) listed in the checkcif report.

2. The chemical formula reported in the cif file is not correct. Why is the formula reported as 0.08(C20 H28 O3)? All the atoms were refined at full occupancy, therefore there is no chemical explanation for the reported formula and sum. This is probably the output of an automatic calculation linked to the wrong Z value used in the res file. The entire molecule consists of C20 H28

O3. The authors must correct the reported formula moiety, sum and formula weight. This would solve the ALERTS G (1,5,6).

3. The dimensions of the measured crystal should be indicated in the cif file. This would solve the ALERTS C (1-3).

4. The centre of gravity is not inside the unit cell. The coordinates of the crystallographic model are indeed completely out of the cell. An additional refinement must be performed to move the molecule inside the cell (MOVE card). This would solve the ALERT C (4).

5. In the cif file the authors reported that the molecule has “_chemical_absolute_configuration s”

The molecule has four stereogenic centres, therefore the absolute configuration should be reported for each asymmetric carbon atom: “_chemical_absolute_configuration 'R (C3), R (C6), R (C9), S (C14)' ”.

The configuration calculated by PLATON (reported below) is consistent with the CIP rules.

PLAT791_ALERT_4_G Model has Chirality at C3 (Chiral SPGR) R Verify

PLAT791_ALERT_4_G Model has Chirality at C6 (Chiral SPGR) R Verify

PLAT791_ALERT_4_G Model has Chirality at C9 (Chiral SPGR) R Verify

PLAT791_ALERT_4_G Model has Chirality at C14 (Chiral SPGR) S Verify

The Flack parameter is also in agreement with the submitted configuration of the molecule. Authors could add the Flack parameter value in the table reporting crystallographic data and refinement details to prove that the crystallographic coordinates of the model are consistent with the proposed configuration.

checkCIF/PLATON report

Alert level C

1) PLAT053_ALERT_1_C Minimum Crystal Dimension Missing (or Error) ... Please Check

2) PLAT054_ALERT_1_C Medium Crystal Dimension Missing (or Error) ... Please Check

- 3) PLAT055_ALERT_1_C Maximum Crystal Dimension Missing (or Error) ... Please Check
- 4) PLAT790_ALERT_4_C Centre of Gravity not Within Unit Cell: Resd. # 1 Note C20 H28 O3

Alert level G

- 1) FORMU01_ALERT_1_G There is a discrepancy between the atom counts in the `_chemical_formula_sum` and `_chemical_formula_moiety`. This is usually due to the moiety formula being in the wrong format.

Atom count from `_chemical_formula_sum`: C1.57 H2.2 O0.24

Atom count from `_chemical_formula_moiety`: C1.6 H2.24 O0.24

- 2) CELLZ01_ALERT_1_G Difference between formula and atom_site contents detected.
- 3) CELLZ01_ALERT_1_G WARNING: H atoms missing from atom site list. Is this intentional?

From the CIF: `_cell_formula_units_Z` 51

From the CIF: `_chemical_formula_sum` C1.57 H2.20 O0.24

TEST: Compare cell contents of formula and atom_site data

atom Z*formula cif sites diff

C 80.07 80.00 0.07

H 112.20 112.00 0.20

O 12.24 12.00 0.24

- 4) PLAT007_ALERT_5_G Number of Unrefined Donor-H Atoms 2 Report
- 5) PLAT042_ALERT_1_G Calc. and Reported MoietyFormula Strings Differ Please Check
- 6) PLAT045_ALERT_1_G Calculated and Reported Z Differ by a Factor ... 0.08 Check
- 7) PLAT791_ALERT_4_G Model has Chirality at C3 (Chiral SPGR) R Verify
- 8) PLAT791_ALERT_4_G Model has Chirality at C6 (Chiral SPGR) R Verify
- 9) PLAT791_ALERT_4_G Model has Chirality at C9 (Chiral SPGR) R Verify
- 10) PLAT791_ALERT_4_G Model has Chirality at C14 (Chiral SPGR) S Verify

Below, we detail how we have addressed the concerns of the reviewers:

Reviewer 1# was supportive of the manuscript without revision.

Reviewer 2# was also supportive of the manuscript pending revisions. The main concerns were:

*Q1: Importantly, the rationalization of the stereoselective syntheses of **24** and **32** (TS1-4) must be corrected. Showing the anions as "lobes" does not correctly portray the species which should be enolates. For example, TS1-1 being disfavoured due to the hindrance between the methyl group and the anion is simply incorrect. The steric*

interactions should be with the incoming electrophile and the enolate.

Regarding Q1: We agree with the referee's comments and revised the structures of the TS in Fig. 5 and 6. In accordance with these revisions, the rationalization of the stereoselective syntheses of **24** and **32** was revised as marked with blue color on pages 12 and 14, respectively.

Q2: *English errors should be corrected.*

Regarding Q2: The manuscript was polished by a native English speaker. We also carefully checked the whole manuscript. The corrected points were marked with red color in the revised manuscript.

Reviewer 3# was also supportive of the manuscript pending revisions. The main concerns were:

Q1: *Authors did not provide the checkcif pdf file for the present structure, although it is clearly stated in the guidelines for authors that "Prior to submitting any cif file, the structure factors and structural output must be checked using IUCr's CheckCIF routine at the <http://checkcif.iucr.org/> website. A pdf copy of the output supplied, explaining any A- or B level alerts should be also provided."*

Regarding Q1: We have provided a checkcif PDF file which was refined according to the reviewer's comments from Q2–Q6.

Q2: *The authors reported a Z value of 51 in the cif and res files. A Z value of 51 is not allowed for the orthorhombic space group P212121. In addition, in the Table of crystallographic data (page S45 of supporting information) the correct Z value of 4 is reported. The authors must perform another refinement cycle with the correct Z value. This would also prevent the ALERTS G (2-3) listed in the checkcif report.*

Q3: *The chemical formula reported in the cif file is not correct. Why is the formula reported as 0.08(C₂₀ H₂₈ O₃)? All the atoms were refined at full occupancy, therefore there is no chemical explanation for the reported formula and sum. This is probably the output of an automatic calculation linked to the wrong Z value used in the res file. The entire molecule consists of C₂₀ H₂₈ O₃. The authors must correct the reported formula moiety, sum and formula weight. This would solve the ALERTS G (1,5,6).*

Q4: *The dimensions of the measured crystal should be indicated in the cif file. This would solve the ALERTS C (1-3).*

Q5: *The centre of gravity is not inside the unit cell. The coordinates of the crystallographic model are indeed completely out of the cell. An additional refinement must be performed to move the molecule inside the cell (MOVE card). This would solve the ALERT C (4).*

Q6: *In the cif file the authors reported that the molecule has “_chemical_absolute_configuration s”. The molecule has four stereogenic centres, therefore the absolute configuration should be reported for each asymmetric carbon atom: “_chemical_absolute_configuration 'R (C3), R (C6), R (C9), S (C14)' ”.*

The configuration calculated by PLATON (reported below) is consistent with the CIP rules.

PLAT791_ALERT_4_G Model has Chirality at C3 (Chiral SPGR) R Verify

PLAT791_ALERT_4_G Model has Chirality at C6 (Chiral SPGR) R Verify

PLAT791_ALERT_4_G Model has Chirality at C9 (Chiral SPGR) R Verify

PLAT791_ALERT_4_G Model has Chirality at C14 (Chiral SPGR) S Verify

The Flack parameter is also in agreement with the submitted configuration of the molecule. Authors could add the Flack parameter value in the table reporting crystallographic data and refinement details to prove that the crystallographic coordinates of the model are consistent with the proposed configuration.

Regarding Q1–Q6: All the errors were corrected and further refinement was performed point-by-point according to the reviewer’s comments. The corresponding alerts were solved after the correction and refinement. In addition, crystallographic data and refinement details were provided in the revised supporting information (see section 5 in SI for details).

Reviewer #2 (Remarks to the Author):

1. The authors have responded to the representation of the enolate structures in TS-1 and TS-2. However, there is still something wrong with them--the authors should build a model and try to view and draw the structures more correctly from this perspective. For example, the enolate TS-1 (Figure 5) and the related structures should show an sp²-hybridized carbon bonded to the oxyanion, yet the structure does not look 180 degrees or flat. Right now it looks 90 degrees. This also goes for the methoxy group in this structure. In Figure 6, modes 1 and 2 for TS-2 shows electron-movement with curly red arrows but the last arrow to show bond formation with E start from the atom and do not start from electrons.

2. The English has improved but errors still remain:

line 103 MHMDS should be LHMDS?

line 128 highly should be high

line 165 should be proved

further proofreading is recommended prior to publication.

Reviewer #3 (Remarks to the Author):

All the points raised in the previous round of review have been satisfactorily addressed.

The authors revised the cif file and provided the checkcif report, as requested.

The manuscript is now recommended for publication.

Dr. Giovanni Bottari
Associate Editor of Nat. Commun.

Manuscript number: NCOMMS-18-05635A

Title: “**Total Synthesis of Cyrneines A–B and Glaucopine C**”

Dear Dr. Bottari:

Thank you and the reviewers for providing valuable comments to the above mentioned manuscript. We have revised the manuscript according to the comments provided by the reviewers. Changes made in the revised manuscript were marked using track changes feature. A detailed response for how we have addressed the referees’ comments was attached at the bottom of this letter. It will be greatly appreciated if you are satisfied with the revised manuscript.

Thank you very much for your great editorship.

Sincerely, yours

F.-S. Han, Dr.

Apr. 14, 2018

Changchun Institute of Applied Chemistry, Chinese Academy of Sciences

5625 Renmin Street, Changchun, Jilin 130022 (China)

Tel: (+) 86-431-8526-2936

E-mail: fshan@ciac.ac.cn

Below, we detail how we have addressed the comments provided the referees.

Reviewer 2# was supportive pending minor revision. The main concerns were:

Q1. There is still something wrong with structures **TS-1** and **TS-2**--the authors should build a model and try to view and draw the structures more correctly from this perspective. For example, the enolate **TS-1** (Figure 5) and the related structures should show an sp²-hybridized carbon bonded to the oxyanion, yet the structure does not look 180 degrees or flat. Right now it looks 90 degrees. This also goes for the methoxy group in this structure. In Figure 6, modes 1 and 2 for **TS-2** shows electron-movement with curly red arrows but the last arrow to show bond formation

with E start from the atom and do not start from electrons.

Regarding Q1: We revised the structures **TS-1** (Fig. 5), **TS-2** (Fig. 6), and the related structures according the referee's comments. For clarity, we drew two work models in Fig. 5 (mode **I** vs mode **II**) instead of one in the previous version. The models in Fig. 6 was re-numbered as mode **III** and mode **IV** accordingly.

Q2. The English has improved but errors still remain: line 103 MHMDS should be LHMDS? Line 128 highly should be high; line 165 should be proved. Further proofreading is recommended prior to publication.

Regarding Q2: We corrected these errors one-by-one. In addition, the whole manuscript was further polished.

Reviewer 3# was supportive of the manuscript without revision.